# A Near-Infrared Fluorescent and Photoacoustic Probe for Visualizing Biothiols Dynamics in Tumor and Liver

**DOI:** 10.3390/molecules28052229

**Published:** 2023-02-27

**Authors:** Weizhong Ding, Shankun Yao, Yuncong Chen, Yanping Wu, Yaheng Li, Weijiang He, Zijian Guo

**Affiliations:** 1State Key Laboratory of Coordination Chemistry, School of Chemistry and Chemical Engineering, Nanjing University, Nanjing 210023, China; 2Chemistry and Biomedicine Innovation Center (ChemBIC), Nanjing University, Nanjing 210023, China; 3Nanchuang (Jiangsu) Institute of Chemistry and Health, Nanjing 210000, China

**Keywords:** fluorescence, photoacoustic imaging, near-infrared probe, biothiols, liver, tumor

## Abstract

Biothiols, including glutathione (GSH), homocysteine (Hcy) and cysteine (Cys), play crucial roles in various physiological processes. Though an array of fluorescent probes have been designed to visualize biothiols in living organisms, few one-for-all imaging agents for sensing biothiols with fluorescence and photoacoustic imaging capabilities have been reported, since instructions for synchronously enabling and balancing every optical imaging efficacy are deficient. Herein, a new near-infrared thioxanthene-hemicyanine dye (Cy-DNBS) has been constructed for fluorescence and photoacoustic imaging of biothiols in vitro and in vivo. Upon treatment with biothiols, the absorption peak of Cy-DNBS shifted from 592 nm to 726 nm, resulting in a strong NIR absorption as well as a subsequent turn-on PA signal. Meanwhile, the fluorescence intensity increased instantaneously at 762 nm. Then, Cy-DNBS was successfully utilized for imaging endogenous and exogenous biothiols in HepG2 cells and mice. In particular, Cy-DNBS was employed for tracking biothiols upregulation in the liver of mice triggered by S-adenosyl methionine by means of fluorescent and photoacoustic imaging methods. We expect that Cy-DNBS serves as an appealing candidate for deciphering biothiols-related physiological and pathological processes.

## 1. Introduction

Sulfhydryl-containing amino acids, such as cysteine (Cys), homocysteine (Hcy) and glutathione (GSH) are bound up with many diseases [1,2,3]. For example, malfunction of Cys level has been associated with hair depigmentation, slowed growth rate and so on. In particular, the abnormal concentration of Cys in the tumor site is significantly higher than that in normal tissues, and it can even reflect different stages of tumor progression [4,5]. An anomalous level of Hcy gives rise to cardiovascular and Alzheimer’s disease [6,7]. Alterations in GSH homeostasis have profound impacts on cell physiology, and it has been demonstrated that GSH deficiency is linked with cancer, liver damage and neurodegenerative diseases [8,9,10]. Additionally, targeting glutathione metabolic pathways has been extensively studied [2,11,12]. For instance, S-Adenosyl methionine (SAM), an intermediate metabolite of methionine, is used for the treatment of depression, liver disorders and osteoarthritis [13,14,15,16,17]. Moreover, the metabolites of SAM, such as Hcy, GSH and decarboxylated SAM, exert vital influences on the function of liver [18,19,20,21]. Unfortunately, the contents of SAM are significantly reduced in liver diseases. To maintain normal liver function, exogenous SAM must be supplemented to compensate for its deficiency, which is beneficial to various tissues, especially the liver [22,23,24]. Thus, tracking the levels of biothiols in the liver is highly conducive for investigating the liver-protective effects of drugs such as SAM.

Owing to their significant roles, efforts to design efficient and handy methods for detecting biothiols with high sensitivity and selectivity are still advancing, which contributes to the understanding of cellular biothiols-related physiopathology [25,26,27,28,29,30]. Among these approaches, optical imaging, such as near-infrared fluorescence imaging (NIRFI), enables non-invasive and real-time visualization of disease-relevant biomarkers with high sensitivity and specificity, drawing great attention [31,32,33,34,35,36,37]. However, the penetration depth of NIRFI is usually limited, making it inaccessible for visualizing bioactive species in deep tissue. In contrast, photoacoustic imaging (PAI) is a burgeoning molecular imaging technique that utilizes the PA effect to convert absorbed light energy into ultrasound signals, which assists researchers to profile deeper tissues with high spatial resolution [38,39,40]. Hence, the drawbacks of NIRFI can be offset by PAI, bringing out the best in each other. Therefore, NIRFI/PAI with complementary advantages of fluorescence and photoacoustic imaging, is expected to perfect imaging output of biomarkers, which endows the ability to self-calibrate to avoid the false-positive signals and systematic errors [41,42,43,44]. Nevertheless, to date, the combination of NIRFI and PAI for multiscale analysis of biomarkers has not yet been fully exploited. Recently, thioxanthene-hemicyanine dye reported by Liang’s group provided an energy balance strategy by transforming existing hemicyanine dyes into thioxanthene-hemicyanine scaffolds, permitting reliable dual modality NIRFI/PAI of biomolecules of interest [45]. We envisioned that an all-in-one imaging agent based on thioxanthene-hemicyanine scaffold for imaging biothiols is highly desirable.

Hemicyanine dyes that display intramolecular charge transfer (ICT) characteristic are important building blocks for advanced optical probes. The ICT process depends on both donor and acceptor strength. When a strong electron-withdrawing group was introduced into the hydroxyl group of the hemicyanine fluorophore, the electron-donating ability of the donor group was weakened, which hampers the charge transfer in the fluorophore. Thus, the fluorescence of the probe could be quenched through blocking ICT process [46,47]. With all this in mind, we constructed a biothiols-activable probe Cy-DNBS with DNBS. Upon exposure to biothiols, the 2,4-dinitrophenylsulfonyl (DNBS) moiety was specifically cleaved and then Cy-DNBS was converted into Cy-OH, resulting in a strong NIR absorption at 726 nm accompanied by the enhanced fluorescence signal at 762 nm. Subsequently, Cy-DNBS was utilized for NIRFI/PAI of endogenous and exogenous biothiols in living cells and mice. Notably, Cy-DNBS have been employed successfully for real-time visualizing alterations in biothiols levels in liver induced by administration of SAM. Taken together, with complementary merits of fluorescence and photoacoustic imaging, Cy-DNBS was successfully applied for dynamically monitoring biothiols in vitro and in vivo, suggesting its enormous potential in imaging biothiols.

## 2. Results and Discussion

### 2.1. Design and Synthesis of Cy-DNBS

We first considered the leaving moiety of the probe since small biomolecules had a limited lifetime in biological system, which called for cleaved groups with excellent electron-withdrawing ability to translate biomarkers into optical signals rapidly. Given that, the first choice of masking group was 2,4-dinitrophenylsulfonyl group (DNBS), which has often been chosen as the responsive unit for biothiols [48,49,50,51]. Therefore, by modifying the hydroxyl group in the fluorophore, we envisioned that the fluorescence could be totally quenched through blocking intramolecular charge transfer (ICT) process, thus minimizing background fluorescence interference. Then, by covalently connecting DNBS to fluorophore, we designed and constructed Cy-DNBS. The reaction of biothiols and Cy-DNBS generated Cy-OH and thereby restored the strong absorption and emission at 726 nm, 762 nm, respectively. The detailed synthetic route and recognition mechanism were shown in Figure 1. Furthermore, the structures of Cy7, Cy-OH and Cy-DNBS were fully characterized by ^1^H NMR, ^13^C NMR and high-resolution mass spectra (HRMS), with the details compiled in the Appendix A. 

### 2.2. Optical Properties of Cy-DNBS towards Biothiols

Before exploring the feasibility of Cy-DNBS for sensing biothiols in vitro and in vivo, the response behaviors of Cy-DNBS to biothiols were investigated. The absorption spectra of Cy-DNBS were first studied after three biothiols were added to PBS buffer (pH 7.4). As depicted in Figure 1a–c, free Cy-DNBS exhibited a maximum absorption peak at 592 nm (ε = 3.33 × 10^4^ M^−1^cm^−1^) and possessed excellent photostability (Appendix A). However, upon the addition of three biothiols, the absorption peak at 592 nm attenuated sharply, and a new strong absorption peak appeared at around 726 nm (ε = 3.42 × 10^4^ M^−1^cm^−1^), resulting from the departure of DNBS moiety and the restored ICT process. To further elaborate on the shift in maximum absorption peak and the sensing mechanism of Cy-DNBS towards biothiols, HRMS and high-performance liquid chromatography (HPLC) analysis were performed to elucidate the reaction processes. The peaks of *m*/*z* 452.20277 and *m*/*z* 682.16604 (Appendix A) were assigned to Cy-OH and Cy-DNBS, respectively, which was consistent with the mechanism proposed by Chakrapani et al. [52]. In addition, the HPLC analysis (Appendix A) also supported the results as stated above, and the retention time of Cy-OH and Cy-DNBS were 20 min and 21.5 min, respectively. Since the major absorption of Cy-DNBS was in the range of NIR, we validated the capability of Cy-DNBS for PAI. Similarly, to the absorption spectra, as shown in Figure 1d–f, an unspectacular PA signal was observed from free probe. In contrast, upon addition of 2 equivalents of Cys, Hcy and GSH, augmented PA signals positioned at 725 nm occurred, and the PA intensity was 4.3-fold, 3.3-fold and 6.0-fold higher than that of the probe alone, respectively. Taken together, these results indicated that Cy-DNBS appeared to be a promising NIR PAI agent for imaging biothiols. 

Then, the fluorescent responses of Cy-DNBS to a series of concentrations of biothiols were investigated. As illustrated in Figure 2a–d, the fluorescence spectra showed that Cy-DNBS displayed an augmented fluorescence peak centered at 762 nm and featured concentration-dependent behaviors. From the fluorescence titration curves and regression equation, the limit of detection of Cys, Hcy and GSH were measured to be as low as 0.25, 0.04 and 0.13 μM, respectively (Appendix A). Selectivity was a key parameter for the probe, and Cy-DNBS responded specifically to biothiols over other bioactive species needed to be assessed. After incubating Cy-DNBS with other competing amino acids in PBS buffer, negligible fluorescence and PA signals were found from interfering species compared to three biothiols, hinting its outstanding selectivity (Figure 2e, Appendix A).

To further explore the response of Cy-DNBS to biothiols following the change in time, the fluorescence spectra were immediately acquired upon Cy-DNBS mixed with three biothiols. As illustrated in Figure 2f, fluorescence intensity of Cy-DNBS increased rapidly and reached a plateau within 100 s. The results indicated that Cy-DNBS could fulfill the demand of rapid identification of biothiols, which was beneficial to capture the variation of biothiols quickly. In addition, it was worth noting that the fluorescent quantum yield of Cy-OH was 0.04 in DMSO by using ICG (Φ = 0.13 in DMSO) as a reference. Fortunately, the low-fluorescence quantum yield promoted the outputs of photoacoustic signals, ensuring a balanced platform for bimodal imaging. These impressive properties encouraged us to explore applications of Cy-DNBS in cells.

### 2.3. Intracellular Biothiols Imaging with Cy-DNBS

Prior to imaging biothiols in cells, we evaluated the biocompatibility of Cy-DNBS in HepG2 cells through conventional MTT assay. It can be seen from Appendix A that no obvious cytotoxicity was observed within the tested concentration range. Following on from this, we enquired the locating tendency of Cy-DNBS in organelles. After Cy-DNBS were co-incubated with three commercial organelle dyes (Mito-, Lyso-Tracker and DAPI) for 30 min, the confocal images were acquired. The results (Appendix A) illustrated that Cy-DNBS were successfully uptaken by cells and prone to remain in the mitochondria and lysosome, with Pearson correlation coefficients (PCC) of 0.74, 0.76 and 0.29 for mitochondria, lysosome and nuclei, respectively. The results were exciting because biothiols are abundant in the two organelles and remarkable indicators of organelles’ function [53,54]. We next examined whether Cy-DNBS could be used to image intracellular biothiols. Firstly, N-Ethylmaleimide (NEM), a thiol-specific scavenger, was used to reduce intracellular biothiols prior to incubation of cells with Cy-DNBS [55]. As displayed in Figure 3a, the cells treated with NEM exhibited much weaker fluorescence than untreated cells. Then, exogenous Cys, Hcy and GSH were added to improve intracellular biothiols content. As expected, fluorescence intensity of cells significantly enhanced compared to Cy-DNBS alone (Figure 3b and Appendix A). All the results showed that Cy-DNBS was suitable for real-time visualization of endogenous biothiols fluctuations in living cells and could be used for a follow-up survey of biothiols in vivo.

### 2.4. Dynamically Tracking Biothiols in Tumor by NIRFI and PAI

Given the probe’s excellent performance in biothiols detection in HepG2 cells, we further studied the applications of Cy-DNBS for tracking biothiols-related dynamic processes in the tumor of a mouse model. By using an excitation wavelength of 740 nm and emission wavelength of 790 ± 20 nm, the interference of spontaneous fluorescence was effectively avoided. After Cy-DNBS (100 μL, 50 μM) was injected into the tumor (tumor group) and normal tissue (control group), respectively, images were recorded immediately. Fluorescence signals increased significantly in both groups within one minute and fluorescence intensity in tumor tissues was nearly four times that of normal tissue (Figure 4a,b). The result could be explained by the reported truth that cancer cells possess higher biothiols content than normal tissues [11,56]. In addition, there were no obvious changes in fluorescence intensity of tumor and normal tissue after 2 min, indicating that Cy-DNBS has reached equilibrium with endogenous biothiols. These results demonstrated that Cy-DNBS could respond quickly to biothiols and differentiate tumor from normal tissues and therefore provided guidance for early diagnosis of tumor progression.

Then, we employed Cy-DNBS for PAI of biothiols in a tumor-bearing mouse model. As depicted in Figure 4c,d, upon injection with Cy-DNBS in two different groups (tumor, normal tissue), respectively, almost no obvious PA signals were observed in normal tissue over time, while strong PA signals were found in tumor, and the intensity was about four times as strong as that of normal tissue. Interestingly, both the fluorescence and photoacoustic intensity of tumor tissue were four times that of normal tissue, and the two were mutually compatible, indicating the excellent authenticity of the imaging outputs. 

PAI possessed the benefits of high resolution and deep imaging depth, which enabled us not only to detect biothiols in tumor, but also to precisely assess the content of biothiols in different tumor regions through capturing numerous two-dimensional tomographic images. It can be seen from Figure 4e that PA intensity was very weak on the tumor surface and increased sharply as the tumor depth deepened, suggesting that there was a highly heterogeneous distribution of biothiols in solid tumors. By contrast, in normal tissues, PA signals hardly change with tissue depth, indicating that normal tissues had no characteristics of distribution differences. These results suggested that PAI provided us with a toolkit for accurately locating and mapping the distribution and abundance of bioactive species at different depths of tumor.

### 2.5. Real-Time Visualizing Biothiols in Liver of Drug-Treated Mouse Model

Liver is vital for digesting food and getting rid of toxic substances, and its dysfunction can lead to a series of diseases. It is well confirmed that the treatment of SAM could be effective against liver disease; hence, we are determined to select SAM as a model drug for visualizing biothiols kinetics in opposing liver diseases [18,20,21]. To prove the feasibility of Cy-DNBS for detecting excess biothiols caused by SAM treatment, we established an SAM-treated mouse model and subcutaneously injected SAM (200 μL, 100 mg/kg) every other day for two weeks. After post injection of Cy-DNBS via tail vein, the fluorescence signals of PBS and SAM group were monitored. As shown in Figure 5a,b, it is clear that mouse administrated with SAM exhibited significantly brighter fluorescence in the liver than that of PBS-treated mouse. Meanwhile, more biothiols were directly found in the liver of the SAM group based on the fluorescence intensities of main organs resected from the mice at 10 min post injection (Figure 5c,d). To further investigate the biothiols fluctuations induced by administration of SAM, the PAI was carried out. The intensity sharply enhanced over time and reached its plateau at 5 min post injection (Figure 5e,f). Noticeably, only when mice were incubated with SAM could we observe apparent augmented signals from the acquired PA images. These results implied that Cy-DNBS could be used for tracking biothiols dynamics in the liver during drug therapy and thus provided feedback on the effects of medicine.

## 3. Conclusions

In summary, a near-infrared fluorescent and photoacoustic probe Cy-DNBS was designed and synthesized by covalently linking thioxanthene-hemicyanine fluorophore with a biothiols-responsive unit (DNBS). Cy-DNBS could selectively respond to biothiols and realized sensitive detection of endogenous and exogenous biothiols in HepG2 cells. In addition, Cy-DNBS could fulfill visualization of biothiols in tumor and differentiate tumor region from normal tissues with high confidence. Furthermore, with the complementary advantages of NIRFI and PAI, Cy-DNBS was successfully applied to precisely monitor the fluctuations of biothiols in the liver triggered by administration of SAM. Accordingly, the dual-modality imaging probe can report the treatment efficacy of drugs that work by regulating the content of biothiols, which may offer a molecular tool for drug screening.

## 4. Materials and Methods

All chemicals used in the synthesis of Cy-DNBS were purchased commercially and used directly without other purification. The intermediates and probe were purified by column chromatography. Silica gel was shipped from Qingdao Haiyang Chemical Co., Ltd. (China), and the size of silica gel particle was around 300–400 mesh. All solvents used for column chromatography were distillation grade. HPLC was carried out on Thermo Scientific Dionex Ultimate 3000 with acetonitrile/H_2_O as eluents. All ^1^H NMR and ^13^C NMR spectra were acquired at 298 K in CDCl_3_ or CD_3_OD on Bruker DRX-400, using TMS as internal reference. The ESI-MS spectra were obtained through Thermo Fisher Scientific LCQ Fleet electron-spray mass spectrometer with positive or negative mode. The HR-MS spectra were presented in positive mode unless otherwise noted. Absorption and fluorescence spectra were recorded using Perkin-Elmer Lambda 35 spectrophotometer and FluoroMax-4 spectrofluorometer, respectively. All the spectroscopic experiments were performed in 3 mL PBS buffer (pH 7.40) by applying a 1 cm optical path standard quartz cuvette as a container. Additionally, the spectral data were processed by Origin 2019 software. The melting point was measured by SGW^®^X-4 melting point apparatus. In vivo imaging of mice was carried out on IVIS Lumina K Series III instrument (PerkinElmer), and isoflurane for anesthetizing mice was purchased from RWD Life Science. The acquired raw images of mice were studied with Living Image 4.5. Confocal imaging was performed with Leica SP8 STED 3X and Zeiss LSM710 microscope. The PCC was determined by one-to-one-pixel matching method in ImageJ software.

### 4.1. Synthesis and Characterization

#### 4.1.1. Synthesis of Indolenine Salt (InS) 

To a solution of 5-chloro-1-pentyne (2.3 mL, 14.2 mmol, 1 eq) in 25 mL acetonitrile, was added potassium iodide (4.71 g, 28.4 mmol, 2 eq). After stirring at 50 °C for 30 min, 2,3,3-trimethylindoles (2.26 g, 14.2 mmol, 1 eq) were added dropwise to the mixture, and then the mixture was heated to reflux. After the reaction completed, the flask was cooled to room temperature, and the inorganic salt was removed by filtration, washed by DCM for several times and used in the next step without further purification (2.51 g, 50% yield. Mp: 189–191 °C, R_f_ = 0.74 (dichloromethane/methanol = 20:1)). ESI-MS calculated 226.15902 for C_16_H_20_N^+^, [M-I]^+^; found m/z: 226.15829.

#### 4.1.2. Synthesis of Cy7

Compound InS (2.51 g, 7.11 mmol, 2 eq), trans-2-chloro-3-(hydroxymethyl) cyclohexyl-1-ene-1-formaldehyde (3.68 g, 3.56 mmol, 1 eq) and sodium acetate (0.59 g, 7.11 mmol, 2 eq) in 100 mL acetic anhydride were heated to reflux under nitrogen atmosphere for 3 h. After the flask was cooled to room temperature, the mixture was poured into 300 mL of cold diethyl ether. The precipitated yellow-green solid was washed with cold diethyl ether, and dried to afford light green powder (1.9 g, 74% yield. Mp: 231–234 °C, R_f_ = 0.48 (dichloromethane/methanol = 20:1)). ^1^H NMR (400 MHz, CDCl_3_) δ 8.37 (d, J = 14.1 Hz, 2H), 7.47–7.39 (m, 6H), 7.33 (s, 2H), 6.35 (d, J = 14.1 Hz, 2H), 4.37 (t, J = 7.3 Hz, 4H), 2.77 (t, J = 6.1 Hz, 4H), 2.47 (td, J = 6.6, 2.5 Hz, 4H), 2.19–2.02 (m, 6H), 2.01–1.92 (m, 2H), 1.76 (d, J = 21.4 Hz, 12H); ^13^C NMR (101 MHz, CDCl_3_) δ 172.5, 150.7, 144.6, 142.1, 140.9, 128.9, 127.8, 125.4, 122.2, 111.1, 101.5, 82.9, 70.2, 49.3, 43.4, 28.2, 26.9, 26.0, 20.6, 16.2. ESI-MS calculated 587.31875 for C_40_H_44_ClN_2_^+^, [M-I]^+^; found m/z: 587.31747.

#### 4.1.3. Synthesis of Cy-OH

3-Hydroxythiophenol (0.27 g, 2.14 mmol, 1.5 eq) and potassium carbonate (0.29 g, 2.10 mmol, 1.5 eq) were dissolved in 15 mL acetonitrile, and the mixture was stirred at room temperature for 20 min. Then, 10 mL Cy7 (1 g, 1.4 mmol, 1 eq) in acetonitrile was added dropwise into the reaction solution, which was heated to 50 °C. After the reaction completed, the solvent was removed by rotary distillation under reduced pressure, and the resulting crude product was purified by column chromatography (CH_2_Cl_2_/CH_3_OH = 50:1, *v*/*v*) to give a blue solid of 0.53 g, with a yield of 65%. Mp: 290–293 °C, R_f_ = 0.63 (dichloromethane/methanol = 20:1). ^1^H NMR (400 MHz, CD_3_OD) δ 8.19 (d, J = 13.5 Hz, 1H), 7.58–7.48 (m, 3H), 7.45–7.38 (m, 1H), 7.33 (d, J = 7.9 Hz, 1H), 7.30–7.24 (m, 1H), 6.96–6.90 (m, 1H), 6.85 (dd, J = 8.8, 2.2 Hz, 1H), 6.31 (d, J = 13.5 Hz, 1H), 4.25 (t, J = 7.4 Hz, 2H), 2.82–2.75 (m, 2H), 2.70 (t, J = 6.1 Hz, 2H), 2.51 (t, J = 2.6 Hz, 1H), 2.38 (td, J = 6.5, 2.6 Hz, 2H), 2.03 (dd, J = 14.1, 6.8 Hz, 2H), 1.93 (dd, J = 11.9, 6.0 Hz, 2H), 1.76 (s, 6H); ^13^C NMR (101 MHz, CD_3_OD) δ 173.4, 171.4, 159.9, 157.1, 143.6, 142.3, 141.4, 141.1, 135.5, 131.2, 129.8, 126.2, 123.8, 123.5, 122.0, 114.6, 113.2, 111.7, 101.0, 83.9, 71.3, 50.4, 43.8, 33.2, 29.0, 27.6, 27.0, 22.0, 16.5. ESI-MS calculated 452.20236 for C_30_H_30_NOS^+^, [M-I]^+^; found m/z: 452.20426.

#### 4.1.4. Synthesis of Cy-DNBS

To a stirred solution of intermediate compound Cy-OH (0.53 g, 0.91 mmol, 1 eq) and triethylamine (0.09 g, 0.91 mmol, 1 eq) in 10 mL CH_2_Cl_2_ under argon atmosphere, 2,4-dinitrobenzenesulfonyl chloride (0.36 g, 1.37 mmol, 1.5 eq) was added quickly. After the reaction finished, the mixture was diluted with CH_2_Cl_2_ and then washed with saturated brine, the organic layer was dried over sodium sulfate anhydrous and concentrated under reduced pressure. The residue was separated by column chromatography (CH_2_Cl_2_/CH_3_OH = 30:1, *v*/*v*) to obtain a blue solid of 0.41 g, yield 48%. Mp: 96–99 °C, R_f_ = 0.33 (dichloromethane/methanol = 20:1). ^1^H NMR (400 MHz, CD_3_OD) δ 8.91 (d, J = 2.2 Hz, 1H), 8.62 (dd, J = 8.7, 2.3 Hz, 1H), 8.50 (d, J = 2.2 Hz, 1H), 8.47–8.40 (m, 2H), 8.32 (d, J = 8.7 Hz, 1H), 8.26 (d, J = 8.6 Hz, 1H), 7.77–7.69 (m, 2H), 7.64–7.50 (m, 4H), 7.20 (dd, J = 8.5, 2.4 Hz, 1H), 7.15 (s, 1H), 6.92 (d, J = 14.8 Hz, 1H), 4.60 (t, J = 7.4 Hz, 2H), 2.84–2.78 (m, 2H), 2.73 (t, J = 6.1 Hz, 2H), 2.56 (t, J = 2.6 Hz, 1H), 2.47–2.39 (m, 2H), 2.18–2.09 (m, 2H), 1.98–1.92 (m, 2H), 1.83 (s, 6H). ^13^C NMR (101 MHz, CD_3_OD) δ 181.2, 153.1, 151.2, 150.1, 147.7, 144.3, 142.5, 137.1, 135.5, 135.1, 133.5, 133.2, 133.2, 132.2, 130.6, 130.5, 129.8, 129.7, 128.0, 126.8, 124.0, 122.6, 122.1, 120.0, 119.7, 115.0, 109.7, 83.5, 71.9, 52.9, 45.9, 33.4, 28.1, 27.9, 27.9, 21.3, 16.4. ESI-MS calculated 682.16762 and 246.96664 for C_36_H_32_N_3_O_7_S_2_^+^ and C_6_H_3_N_2_O_7_S^-^, found m/z: 682.16803 and 246.96629, respectively. 

### 4.2. Detection Methods

#### 4.2.1. Optical Study

Cy-DNBS was dissolved in 1.5 mL dimethyl sulfoxide (DMSO) to prepare 10 mM stock solutions. The stock solution of three biothiols were prepared by dissolving Cys, Hcy and GSH in PBS buffer, respectively. Then, the stock solution was stored in the refrigerator at −20 °C for use. Fluorescence spectra were measured with a 5 nm slit for both excitation and emission. 

#### 4.2.2. Quantum Yield Calculation

The quantum yield was measured by using ICG (Φ = 0.13 in DMSO) as a standard. The following equation was used to calculate quantum yield of Cy-DNBS Φ_x_:Φ_x_ = Φ_s_ (A_s_ F_x_/A_x_ F_s_) (n_x_**/**n_s_) ^2^
where Φ_s_ is the quantum yield of ICG, A is the absorbance at the excitation wavelength, F is the area under the corrected emission curve, and n is the refractive index of the solvents used, respectively. 

#### 4.2.3. MTT Assay

HepG2 cells were cultured in Dulbecco’s Modified Eagle Medium (DMEM, Gibco) supplemented with 10% fetal bovine serum (FBS; Gibco) and glutamine (2 mM) in an atmosphere of 5% CO_2_ and 95% air at 37 °C. The cellular cytotoxicity of Cy-DNBS towards HepG2 cells were evaluated using the classical MTT assay. An amount of 100 μL of DMEM medium containing 1000–10,000 HepG2 cells was seeded into each well of 96-well plate. Then, Cy-DNBS was diluted to different concentrations with culture medium and gradually added to each well sequentially according to the concentration gradient. The cells were incubated at 37 °C, 5% CO_2_, for 24 h. Next, 40 μL of MTT solution (2.5 mg/mL) was added to each well, and the cells were incubated for another 4 h. After the original medium were removed, 150 μL of DMSO was added to each well and placed on a shaker for 10 min to allow the crystals to dissolve effectively. Finally, the absorbance value of each well at 490 nm was measured in a microplate reader and repeated three times to calculate cell viability.

#### 4.2.4. Cell Imaging

Firstly, to improve the biothiols content of cells, DMEM medium containing exogenous Cys (2 mL, 4 μM) was incubated with HepG2 cells in the confocal dishes for 30 min. Then, the medium was replaced with Cy-DNBS (2 mL, 4 μM) in DMEM medium. Meanwhile, the cells were incubated with three commercial organelle dyes (Mito-Tracker, Lyso-Tracker and DAPI, 1.5 mL, 1 μM) for 30 min, respectively. At last, the cells were washed with 2 mL PBS buffer 3 times before imaging on laser confocal microscope. Green channel (λ_ex_ = 488 nm, λ_em_ = 500–620 nm), Red channel (λ_ex_ = 633 nm, λ_em_ = 680–800 nm), Blue channel (λ_ex_ = 405 nm, λ_em_ = 420–550 nm).

#### 4.2.5. Imaging of Cy-DNBS towards Endogenous and Exogenous Biothiols

The HepG2 cells were divided into two groups, which were incubated with DMEM medium in the presence and absence of NEM (2 mL, 1 mM) for 30 min. Then, the medium was replaced with DMEM medium containing Cys, Hcy and GSH solution (2 mL, 1 mM), respectively. After culturing the cells for 30 min, the medium was replaced with DMEM medium containing Cy-DNBS (2 mL, 4 μM) and incubated for another 30 min. At last, the cells were washed with PBS buffer 3 times and placed under confocal fluorescence microscope for direct imaging.

#### 4.2.6. In Vivo Fluorescence Imaging

For in vivo fluorescence imaging of biothiols in tumor, mice were anaesthetized using isoflurane. Cy-DNBS (100 µL, 50 µM) was injected into the subcutaneous tumor and normal tissues (control group) of the mice, respectively, and then mice were immediately imaged. As for imaging biothiols in the liver, the mice were divided into two groups, including SAM and PBS Group. Both groups received SAM/PBS (200 μL, 100 mg/kg) intraperitoneally every two days for two weeks. Then, Cy-DNBS (100 µL, 50 µM) was injected by tail vein. After imaging was completed, the mice were sacrificed, and the main organs (heart, liver, spleen, lung and kidney) were harvested and subjected to ex vivo imaging. The fluorescence imaging was performed with a band path of 770–810 nm upon excitation at 740 nm. 

#### 4.2.7. In Vivo Photoacoustic Imaging

For in vivo photoacoustic imaging of biothiols in tumor, isoflurane was used to anaesthetize mice during imaging. Cy-DNBS (100 µL, 50 µM) was injected into the subcutaneous tumor and normal tissues (control group) of the mice, respectively. Then, mice were immediately imaged. Other operating conditions of PA imaging in the liver are similar to that of fluorescence imaging. The excitation wavelength at 725 nm was selected according to the UV-Vis spectra. 

## Data Availability

The data supporting the reported results can be found within the article and its Appendix A.

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
