# Peer review of "A Near-Infrared Fluorescent and Photoacoustic Probe for Visualizing Biothiols Dynamics in Tumor and Liver"

_molecules, 2023, doi:10.3390/molecules28052229_

Round 1

Reviewer 1 Report

In this manuscript, the authors reported that an activatable NIR dye was synthesized and utilized to recognize selectively for biothiols. This probe can be utilized for NIRFI/PAI of endogenous and exogenous biothiols in living cells and mice. Furthermore, this probe was successfully applied to precisely monitor the fluctuations of biothiols in the liver triggered by administration of SAM. The results were interesting and this manuscript could be published after revision.

1. In Section 2.1, the authors claimed that “by modifying the hydroxyl group in the fluorophore, we envisioned that the fluorescence could be totally quenched through blocking intramolecular charge transfer (ICT) process, thus minimizing background fluorescence interference” why modified the hydroxyl group could quench the fluorescence of the probe. The author should explain this more thoroughly.

2. To further validate the sensing mechanism of the probe towards biothiols, a more detailed analysis should be done. For instance, the reaction mixtures of Cy-DNBS with respective biothiols could be subjected to HPLC analysis.

3. In section 2.2, PA images of Cy-DNBS towards other bioactive species needed to be assessed.

4. In Fig.S5 the authors have to explain better how is calculated the Pearson coefficient: the PCC is obtained from the fluorescent of all the figure?

Author Response

Reviewer #1:

In this manuscript, the authors reported that an activatable NIR dye was synthesized and utilized to recognize selectively for biothiols. This probe can be utilized for NIRFI/PAI of endogenous and exogenous biothiols in living cells and mice. Furthermore, this probe was successfully applied to precisely monitor the fluctuations of biothiols in the liver triggered by administration of SAM. The results were interesting and this manuscript could be published after revision.

  1. In Section 2.1, the authors claimed that “by modifying the hydroxyl group in the fluorophore, we envisioned that the fluorescence could be totally quenched through blocking intramolecular charge transfer (ICT) process, thus minimizing background fluorescence interference” why modified the hydroxyl group could quench the fluorescence of the probe. The author should explain this more thoroughly.

Response: Thanks for your suggestion, we have described the details of the design principle and fluorescence quenching mechanism of the probe in “Introduction” of the revised manuscript. The supplemented text are as follows:

       Hemicyanine dyes that display intramolecular charge transfer (ICT) character are important building blocks for advance optical probes. The ICT process depends on both donor and acceptor strength. When strong electron-withdrawing group was introduced into the hydroxyl group of the hemicyanine fluorophore, the electron donating ability of the donor group was weakened, which hampers the charge transfer in the fluorophore. Thus, the fluorescence of the probe could be quenched through blocking ICT process [46,47]. With all this in mind, we constructed a biothiols-activable probe Cy-DNBS. (Page 2)

  1. To further validate the sensing mechanism of the probe towards biothiols, a more detailed analysis should be done. For instance, the reaction mixtures of Cy-DNBS with respective biothiols could be subjected to HPLC analysis.

Response: As reviewer’s suggestion, the reaction mixtures of Cy-DNBS with GSH were subjected to HPLC analysis to further validate the sensing mechanism of the probe towards biothiols. This part was presented in Figure S3 in “Supporting Information” and discussed in the revised manuscript. (Page 4)

Figure S3. HPLC chromatograms of a) Cy-OH, b) Cy-DNBS and c) the mixtures of Cy-DNBS with GSH over 2 min. The mobile phase was acetonitrile and water.

  1. In section 2.2, PA images of Cy-DNBS towards other bioactive species needed to be assessed.

Response: Thanks for pointing this out. We have checked the selectivity of Cy-DNBS towards other bioactive species via PA imaging. This part was presented in Figure S6 in “Supporting Information” and discussed in the revised manuscript. (Page 4)

Figure S6. Histogram of the PA intensity of Cy-DNBS (10 μM) towards Cys, Hcy, GSH and other competing amino acids including Phe, Ser, Trp, Glu, Gly, His, Ala, Arg, Asp (20 μM) at 725 nm in PBS solution (pH 7.40).

  1. In Fig.S5 the authors have to explain better how is calculated the Pearson coefficient: the PCC is obtained from the fluorescent of all the figure?

Response: We used one-to-one-pixel matching method in ImageJ software to calculate PCC. In pixel matching colocalization analysis, the intensity of a pixel in red channel is evaluated against the corresponding pixel in the green channel of a colocalization images, generally producing a scatterplot from which the PCC is determined.

Reviewer 2 Report

Herein, the authors designed a dual-modality imaging probe Cy-DNBS,generated a strong NIR absorption as well as turn-on PA signal after treatment with biothiols. Cy-DNBS was employed for tracking biothiols upregulation in the liver of mice triggered, and displayed promising prospect as the imaging tool to obtain multi-scale information of biomarkers. This work is interesting and the experimental data were well organized, so I recommend this manuscript to be accepted for the publication in Molecules with major revision.

1. The authors should provide the stability performance for the probe? such as photo-stability.

2. How about the solubility of the probe?

3. In the co-localization experiment, why the probe exhibited different fluorescence positions in the red channel.

4. The emission wavelength of the compound is 762 nm, why the receiving wavelength is 700-750 nm. In addition, whether the author could quantify the fluorescence intensity in Figure3. 

5. Error bar should be added in Figure S2 and Figure S3.

6. At present, dual-modal probe for NIR fluorogenic and ratiometric photoacoustic imaging of Cys/Hcy has been reported a lot, so the authors should further explain the innovation of Cy-DNBS. 

Author Response

Reviewer #2:

Herein, the authors designed a dual-modality imaging probe Cy-DNBS,generated a strong NIR absorption as well as turn-on PA signal after treatment with biothiols. Cy-DNBS was employed for tracking biothiols upregulation in the liver of mice triggered, and displayed promising prospect as the imaging tool to obtain multi-scale information of biomarkers. This work is interesting and the experimental data were well organized, so I recommend this manuscript to be accepted for the publication in Molecules with major revision.

  1. The authors should provide the stability performance for the probe? such as photo-stability.

Response: Thanks for your kind suggestion, the photostability of the probe was evaluated by recording the changes of maximal absorbance of the probe within 3min. The result indicated that Cy-DNBS possessed excellent photostability under light irradiation. This part was displayed in Figure S2 and discussed in the revised manuscript. (Page 3)

Figure S2. The photostability of Cy-DNBS (10 μM) in methanol was estimated by irradiating solutions in a quartz cuvette with a 635 nm laser (50 mW/cm2) for 3 min at a 30 second interval.

  1. How about the solubility of the probe?

Response: Thanks a lot for the reviewer’s comment. The probe (10 μM) has good solubility in PBS buffer (pH = 7.4), and all the spectral tests were performed in PBS buffer without visible deposits.

  1. In the co-localization experiment, why the probe exhibited different fluorescence positions in the red channel.

Response: Indeed, as reviewer has proposed, the probe seemed to exhibit different fluorescence positions in the red channel when Cy-DNBS were co-incubated with commercial nuclei dye DAPI. We think that the difference resulted from the weaker brightness of fluorescence compare to its counterparts. And we have subtly tuned the brightness of fluorescence in the red channel (Figure S8 in the revised “Supporting Information”). Thanks again for your valuable comments, which are very important and instructive for our future research.

Figure S8. HepG2 cells were pretreated with exogenous Cys (2 mL, 4 μM) for 30 min and then Cy-DNBS (2 mL, 4 μM) were co-incubated with three commercial organelle dyes (Mito-Tracker, Lyso-Tracker and DAPI, 1 μM) for 30 min. Green channel (λex = 488 nm, λem = 500-620 nm), Red channel (λex = 633 nm, λem = 680-800 nm), Blue channel (λex = 405 nm, λem = 420-550 nm). Scale bar: 50 μm.

  1. The emission wavelength of the compound is 762 nm, why the receiving wavelength is 700-750 nm. In addition, whether the author could quantify the fluorescence intensity in Figure3.

Response: We are very sorry for this mistake, the receiving wavelength in Figure 3 has been changed 700-750 nm to 680-800 nm, and the fluorescence intensity in Figure 3 has been quantified in Figure S9 in the revised “Supporting Information”. 

  1. Error bar should be added in Figure S2 and Figure S3.

Response: We are very sorry for this negligence, the error bar in Figure S2 and Figure S3 (Figure S3 and Figure S4 in the revised SI) has been added in the revised “Supporting Information”.

  1. At present, dual-modal probe for NIR fluorogenic and ratiometric photoacoustic imaging of Cys/Hcy has been reported a lot, so the authors should further explain the innovation of Cy-DNBS.

Response: Thanks for your comment. As for the novelty, although there are some reports focusing on fluorescence and photoacoustic dual modal imaging contrast agents for biothiols, the applications of dual modal imaging method for assessing therapeutic effect of drugs that could convert into biothiols has been rarely explored. S-Adenosyl methionine (SAM), a precursor of Hcy, is used for the treatment of depression, liver disorders, and osteoarthritis. Especially, SAM exerts vital influences on the function of liver. Thus, tracking the level of biothiols in the liver is highly conducive for investigating the liver-protective effects of drugs such as SAM. In this report, Cy-DNBS was successfully applied to precisely monitor the fluctuations of biothiols in the liver triggered by administration of S-adenosyl methionine (SAM) by means of fluorescent and photoacoustic imaging methods. This work provides a molecular tool to assess the treatment efficacy of drugs that work by regulating the content of biothiols. Therefore, we believe that this work is appropriate for publication in Special Issue “Fluorescent Probes: From Structure Design to Property Tuning and Versatile Application” in journal Molecules.

Reviewer 3 Report

In this paper, the authors reported a new near-infrared thioxanthene-hemicyanine dye (Cy-DNBS) for fluorescence and photoacoustic imaging of bio-thiols in vitro and in vivo. As well as, Cy-DNBS was utilized for imaging of endogenous and exogenous bio-thiols in HepG2 cells and mice. However, I recommend acceptance of this manuscript, still some concerns need to be fixed by authors.

Comments

  1. I suggest authors should change the word to new instead of novel in the abstract.

2.     In experimental, Mps should be included if reported one should compare with those as well Rf should be provided.

3.     In SI file, NMR window range should always in b/w 0-10 ppm for 1H and 0-220 ppm for 13C.

4.     I can see that several typos have been appeared in main article and SI file, NMR data rechecked again with provided copies, language needs to be polish accordingly.

5.     The images resolution of some of the figures should be improved as better for understanding the content with respect to readers.

Author Response

Reviewer #3:

In this paper, the authors reported a new near-infrared thioxanthene-hemicyanine dye (Cy-DNBS) for fluorescence and photoacoustic imaging of bio-thiols in vitro and in vivo. As well as, Cy-DNBS was utilized for imaging of endogenous and exogenous bio-thiols in HepG2 cells and mice. However, I recommend acceptance of this manuscript, still some concerns need to be fixed by authors.

  1. I suggest authors should change the word to new instead of novel in the abstract.

Response: Thanks for the suggestion, we have replaced the word “novel” with “new”in the revised manuscript. (Page 1)

  1. In experimental, Mps should be included if reported one should compare with those as well Rf should be provided.

Response: As reviewer’s suggestion, we have measured the melting points and Rfs of compounds and presented these results in the revised “Supporting Information”.

  1. In SI file, NMR window range should always in b/w 0-10 ppm for 1H and 0-220 ppm for 13C.

Response: As reviewer’s suggestion, the NMR window range have adjusted to 0-10 ppm for 1H NMR graphs and 0-220 ppm for 13C NMR graphs, as shown in the revised “Supporting Information”.

  1. I can see that several typos have been appeared in main article and SI file, NMR data rechecked again with provided copies, language needs to be polish accordingly.

Response: Thanks for your comments. The NMR data have rechecked again with provided copies. In addition, we are very sorry for the poor language of our manuscript. We have carefully scrutinized the manuscript and made corresponding revisions including some typos and grammatical errors. We really hope that the readability and language level have been substantially improved.

  1. The images resolution of some of the figures should be improved as better for understanding the content with respect to readers.

Response: Thanks for your kind suggestion, which is valuable for improving the quality and readability of the manuscript. We have tried our best to replace some images with the higher resolution ones to help readers understand the content better.

Round 2

Reviewer 2 Report

With all the revisions and improvements the authors have made, the manuscript could now be accepted for publication.